# From Orange Juice By-Product in the Food Industry to a Functional Ingredient: Application in the Circular Economy

**DOI:** 10.3390/foods9050593

**Published:** 2020-05-06

**Authors:** Larissa Alves de Castro, Jaqueline Miranda Lizi, Eduardo Galvão Leite das Chagas, Rosemary Aparecida de Carvalho, Fernanda Maria Vanin

**Affiliations:** Food Engineering Department, University of São Paulo, Faculty of Animal Science and Food Engineering (USP/FZEA), Laboratory of Bread and Dough Process (LAPROPAMA), Av. Duque de Caxias Norte 225, Pirassununga 13635-900, SP, Brazil; larissa.castro@usp.br (L.A.d.C.); jaqueline.lizi@usp.br (J.M.L.); eduardo.chagas@usp.br (E.G.L.d.C.); rosecarvalho@usp.br (R.A.d.C.)

**Keywords:** dietary fiber, minerals, phenolic compounds, antioxidant properties, environmental waste

## Abstract

In the orange juice industry, more than 50% of raw material becomes by-products that are rich in active compounds and have high nutritional content. Improved use of these by-products could represent a key strategy for a circular economy. The objective of this study was to produce a flour from orange juice by-product, characterize it, and then apply this flour to produce cookies. Orange by-product flour (OBPF) was characterized in terms of its chemical composition, dietary fiber, phenolic compounds, antioxidant potential, and hygroscopic properties. Subsequently, the effect of substituting wheat flour by OBPF in cookies was evaluated. OBPF presented a very high content of dietary fiber (73.61% dry matter (DM)), minerals (ash = 2.72% DM), and total phenolic compounds (534 ± 30 mg gallic acid equivalent (GAE)/100 g of DM). In general, the properties of cookies were not significantly influenced by using OBPF as a substitution for wheat flour. Sensorial analyses showed that cookies produced with 10% OBPF presented the higher scores. Therefore, OBPF showed interesting characteristics, suggesting its possible use in the development of fiber enriched foods such as cookies; and its production represents a key strategy for the orange juice processing industries towards the application of a circular economy in the food system.

## 1. Introduction

In accordance with the Food and Agriculture Organization (FAO) [1], one-third of all food produced in the world is wasted, which represents approximately 1.3 billion tons. Waste products are largely generated by food processes, such as vegetable oil extraction, starch, juice and sugars production, and animal wastes such as bones, offal and hides, and also whey protein from cheese processing. Fruits and vegetables have the highest waste rates of any food, i.e., 45% [2,3], which in developing regions such as Asia, Africa, and Latin America are concentrated in agriculture and processing. Because of how they are processed, a significant amount of these foods is traditionally discarded. When fruit is processed, parts such as the core, peel, pips, and kernel are discarded. Although these by-products still contain nutrients and bioactive compounds, they are considered a problem. Due to the high-water content of these by-products, they are prone to microbial deterioration, and therefore they are commonly used for animal feed or as fertilizer, or they are disposed of in landfills or incinerated. In addition to the inefficient use of resources and high rates of food wasted in all stages of the food system, the world population has increased. Therefore, the need for more sustainable practices, with reduced environmental impacts towards the application of a circular economy in the food system, represents a key strategy for the future. Furthermore, in accordance with recently published information by the FAO considering the impacts of the novel coronavirus (COVID-19), the situation of populations with extreme hunger in the world will probably increase, and therefore countries should gain efficiencies and try to reduce trade-related costs, for example, by reducing food waste and losses [4].

In tropical and subtropical regions, the orange (*Citrus cinensis*) is an important crop. Brazil is the major producer, with 18 million tons in 2017 [5], and accounts for three-quarters of global orange juice exports. United States, India, Mexico, and China are also extensive producers, and there are some other countries, such as Spain and Egypt which annually produce more than two million tons, and Iran, Italy, and Indonesia, with one million ton [5]. More than orange producers, these countries are also becoming processors [6]. Peel (flavedo) and rag (albedo) by-products are generated from the production of citrus juice, and represent about 45%–60% of the fruit [7,8]. In accordance with the USDA reports, in 2019 and 2020 [9], the global orange juice production was estimated to be 1.7 million tons. Considering these numbers, it could be estimated that global orange juice production could generate between 0.8 to 1 million tons of by-products, or waste, each year. Differences between the *Citrus sinensis* cultivars are observed, in relation to the physical and chemical characteristics of the fruits, similarities are found in terms of water content and dry matter [10], and therefore by-products generated by juice process could be considered to be similar.

As previously detailed, orange juice by-product is mostly destined for animal feed [11], or for essential oils [12,13] or carotenoids [14] extraction from peels, as well as other minor uses. However, previous studies have reported higher dietary fiber values from the peel of orange [15] and pomace [16], and also higher levels of phenolic compounds and antioxidant capacity [15] as compared with the fruit. 

A significant number of studies have demonstrated a clinical relation between the reduction of cardiovascular disease levels and other diseases, with an increase of fruit consumption, and this effect has been attributed to their high levels of phenolic compounds and antioxidant properties, and dietary fiber intake [17]. The recommendation of dietary fiber intake is >25 g per day of total dietary fiber [18], however Europe and the United States populations consume only 30% of this value [19]. The physiological effect of dietary fiber intake has been related to a higher proportion of soluble dietary fiber. However, some studies have been shown to be more useful evaluate such properties in relation to the functional properties, as the water absorption index [20].

Consequently, the increased number of studies to improve the ways to extract maximum value from food by-products, add them back into foods, and reduce their environmental impact, represents a key strategy. Furthermore, it is in agreement with the Food and Agriculture Organization of the United Nations (FAO) 2013 [21] recommendations, “to prevent food waste, to reduce its economic and environmental impact, but finding new uses for food products that do end up being discarded”. In addition, fruit groups, with vegetables, are the most responsible for food waste in all supply chain stages, i.e., around 45%, and approximately 21.7% of this food waste is generated at the processing and manufacturing stages [3].

Therefore, the orange juice by-product industry could represent a strategy to elevate the level of “fruit” intake and its associated compounds, resulting in the development of new natural ingredients for the food industry and lower food wastes during process stages, and therefore increased efficiency. For this, an adequate process for developing the by-product and its complete characterization, in terms of composition, and also in relation to functional properties, is necessary before its application in real and easily manufactured food, which are the objectives of this study.

## 2. Materials and Methods

### 2.1. Materials

Sweet oranges (*Citrus sinensis* L. Osbeck) were purchased from the local supermarket (São Paulo, Brazil). Margarine (Doriana, Brazil), brown sugar (Siamar, Brazil), refined sugar (Caravelas, Brazil), wheat flour (Renata, Brazil), and chemical leavener (Dona Benta, Brazil) (a mix of sodium bicarbonate, monocalcium phosphate, and calcium carbonate) were also purchased locally. For the quantification of dietary fiber (DF) a K-TDFR-200a kit from Megazyme International (Wicklow, Ireland) was acquired. The reagents used were methanol (Synth, São Paulo, Brazil), Folin–Ciocalteau reagent (Sigma-Aldrich, Bellefonte, PA, USA), sodium carbonate (Synth, São Paulo, Brazil), gallic acid (Sigma-Aldrich, Bellefonte, PA, USA), ferric chloride (Synth, São Paulo, Brazil), chloridric acid (LS chemicals, Ribeirão Preto, Brazil), (-6-hidroxi-2,5,7,8-tetramethylchroman-2-carboxilic acid (Trolox; Sigma-Aldrich, Bellefonte, PA, USA), acetic acid (Synth, São Paulo, Brazil), sodium acetate (Synth, São Paulo, Brazil), sodium phosphate monobasic (Synth, São Paulo, Brazil), sodium phosphate bibasic (Synth, São Paulo, Brazil), 2,4,6-tripyridyl-s-triazine (TPTZ) (Sigma-Aldrich, Bellefonte, PA, USA), sodium fluorescein salt (Sigma-Aldrich, Bellefonte, PA, USA), and 2,2′-azobis (2-methylpropionamidine) dihydrochloride (AAPH) (Sigma-Aldrich, Bellefonte, PA, USA).

### 2.2. Methods

#### 2.2.1. Orange By-Product Flour Production

The orange by-product flour (OBPF) was produced in accordance with Santana [22] with some adaptations. First, oranges were manually selected, washed, and peeled. The albedo which was obtained after juice extraction was manually triturated and washed in water for 30 min. Then, the material was oven-dried (Marconi, MA035/5, Brazil) at 60 °C for 24 h, ground in a mill (Marconi, MA340, Brazil), and sieved (16 mesh). The resulting OBPF was stored in a refrigerator (7 °C).

#### 2.2.2. Orange By-Product Flour Characterization

##### Chemical Composition

The moisture content, crude protein, crude fat, and ash of the OBPF were determined using the methodology defined by AACC (2000) [23]. The total dietary fiber (DF) was determined according to the methodology defined by AOAC (1985) [24].

##### Flour Extraction

The extraction for the analysis of total phenolic and antioxidant potential was performed using a hydroalcoholic solution (methanol, 80:20 *v*/*v*). The sample (1 g) was added in 100 mL of solvent and homogenized (1 min, 6000 rpm) with ultra-turrax (IKA, T25 digital, Germany). Then, the sample was vacuum filtered (Tecnal, TE058, Brazil), using a paper filter (Whatman grade 1). The retained material was extracted again two more times, however, with only 50 mL of the solvent in each step. Extractions were prepared in triplicate and stored at −22 °C.

##### Total Phenolic Content

The total phenolic (TP) content of the OBPF was determined in accordance with Singleton et al. [25]. Sample extracts (0.5 mL) were added to 2.5 mL of Folin–Ciocalteau reagent in tubes and allowed to stand for 4 min, and then 2 mL of sodium carbonate (7.5%) was added to the tube and homogenized using a Vortex agitator (IKA, Vortex 1V1, Germany). The tubes were incubated for 2 h (25 ± 2 °C) in the absence of light. The sample absorbance was measured in a wavelength of 740 nm using a spectrophotometer (PerkinElmer, Lambda 35, Shelton, CT, USA). Gallic acid was used as the standard and the results were expressed as mg of gallic acid equivalents (GAE)/100 g of dry matter.

##### Antioxidant Potential

FRAP (Ferric Reducing Antioxidant Power) Assay

The FRAP assay was performed according to methodology described by Benzie and Strain [26] and 2.9 mL of FRAP solution (acetate buffer (300 mM); 2,4,6-tripiridil-S-triazina solution (10 mM); ferric chloride (20 mM); in a ratio of 10:1:1), was homogenized with 0.1 mL of extract. The mixture was kept at 37 °C for 30 min in a thermostat bath (Marconi, MA 159, Brazil). Sample absorbance was measured in a wavelength of 593 nm in a spectrophotometer (PerkinElmer, Lambda 35, USA). Trolox was used as an external standard and the results were reported as µmol of trolox equivalent (TE)/100 g of dry matter.

ORAC (Oxygen Radical Absorbance Capacity) Assay

The ORAC assay was used to determine the antioxidant potential of the OBPF [27]. In a microplate, an aliquot (150 µL) of fluorescein solution (81 mM) and 25 µL of extracts were added to the cells. The microplate was incubated at 37 °C for 10 min in a spectrofluorimeter (BMG Labtech, FLUOstarOPTIMA, German). Then, 25 µL of 2,2-azobis (2-methylpropionamidino) dihydrochloride (152 mM) was added to each cell and the determination of the decay of fluorescein was performed with 485 nm excitation wavelength and 528 nm emission performed every 1 min for 120 min. Trolox was used as the external standard and the results were reported as mg of trolox equivalent (TE)/100 g of dry matter.

##### Phenolic Profile

The phenolic compounds identification was performed according to Hassimoto et al. [28], using a 1260 InfinitQuatrary quaternary LC system (Agilent Technologies, Santa Clara, CA, USA), consisting of an autosampler and a quaternary pump coupled to a diode array detector (DAD). Using the Poroshell 2.7 µ (100 × 3 mm) column (Agilent Technologies, Santa Clara, CA, USA) with the following parameters: 0.5 mL/min flow rate and 25 °C temperature. The following two solvents were used in the mobile phase: 0.5% formic acid in water and acetonitrile. For analysis, the solvent gradient was applied as follows: 5%–18% acetonitrile at 0–7 min, 18%–28% at 7–17 min, 90% at 17.01–20 min, 5% at 21–26 min. For detection of anthocyanins, elution was monitored at 525 nm and the other flavonoids were monitored at 270 and 370 nm. LC-QTOF-MS/MS analyses were performed using a Prominence liquid chromatography (Shimadzu, Japan) coupled with an Esquires-LC ion trap mass spectrometer (Bruker Daltonics, Billerica, MA, USA) using an ionization interface electrospray (ESI) and the same solvent gradient was used in the HPLC-DAD, reducing the flow rate to 0.2 mL/min. A positive mode ESI was used to identify anthocyanins and a negative mode was used for the other flavonoid classes. The parameters required for mass spectrometer operation were as follows: collision energy at 4500 V positive mode and 4000 V negative mode and the capillary temperature was 275 °C. In order to identify the compounds, the data obtained were compared with the retention times of commercial standards when possible, also by absorption spectrum similarity, mass spectral characteristics, and comparison of literature data.

##### Hydration Properties

The water absorption index (WAI) and the water solubility index (WSI) of the OBPF were determined. For the WAI, the sample (1 g) was diluted in 25 mL of distilled water, using a centrifuge tube. The tubes were shaken for 30 min and placed in a centrifuge (Eppendorf, Centrifuge 5430-R, Hamburg, Germany) at 2500 rpm for 10 min. The supernatant meal contents were placed in a petri dish and the material contained in the tube was weighed. The test was performed in triplicate. The WAI calculation was based on the OBPF mass ratio hydrated by the dehydrated OBPF mass, obtaining the result in g water/g dry matter. For the WSI, the supernatant of WAI was placed in petri dishes and oven-dried (FANEM, 515, Brazil) for 15 h at 105 °C. The analysis was performed in triplicate, and the result, expressed as a percentage, was calculated by the ratio of the mass of the dehydrated solid to the mass of the dehydrated flour.

For the oil absorption index (OAI), 4 g of the OBPF and 24 mL of soybean oil were placed in 50 mL centrifuge tubes, following the methodology of Sosulski et al. [29]. The tubes were shaken for 30 min, followed by centrifugation at 2500 rpm for 25 min. The supernatant was discarded, and the analysis was repeated in triplicate. To calculate the OAI, the mass of the insoluble residue was divided by the initial OBPF mass, and the result was obtained in g oil/g dry matter.

#### 2.2.3. Cookie Production

The methodology adapted from Pareyt and Delcour [30] was used for the production of the cookies. Control cookies were produced using 238.5 g of wheat flour, 43.5 g of margarine, 86 g of refined sugar, 86 g of brown sugar, 5 g of chemical leavener, and 41 g of water (Table 1). First, margarine and sugar were mixed for 3 min in an electric mixer (KitchenAid, Stand Mixer Pro Line 6.9 L, Greenville, OH, USA). Then, the water was added and mixed for two more minutes. Finally, the wheat flour and chemical leavener were added and mixed for two more minutes.

Thereafter, the obtained dough was handled and molded using a cutter (∅ = 5 mm) for cookies of approximately 16 g. The cookies were placed in a rectangular baking dish and baked in a preheated industrial oven (Klimaquip CF-20, Campinas, SP, Brazil) for 7 min at 165 °C. Preliminary tests were performed to verify maximal levels of OBPF substitution. Cookies produced with OBPF were produced by wheat flour substitution using the OBPF at concentrations of 5%, 10%, 15%, and 20%, using the same protocol.

#### 2.2.4. Cookie Characterization

##### Physical Properties

The baked cookies were measured in relation to their diameter (D) and height (H) using a micrometer. Subsequently, by dividing D by H the spread factor (SF) of the cookie was determined [31]. Six cookies were taken from each recipe and measured. Each recipe was made in triplicate, and therefore a total of 9 values were obtained for each formulation.

##### Specific Volume

Cookies were first weighed using a semi-analytical balance (Shimadzu, São Paulo, SP, Brazil) and the apparent volume determined by using a VolScan profile (VSP300, Stable Micro Systems, Godalming, UK) immediately after the end of baking. The specific volume was calculated by dividing the apparent volume by the weight of sample. Three cookies were taken from each recipe and tested. Each recipe was made in triplicate, and therefore a total of 9 values were obtained for each formulation.

##### Hardness

A texture analyzer TA–XT2 Plus (Texture Analyzer) (Stable Microsystems SMD, Godalming, UK) equipped with a 5 kg load cell was used to measure the hardness of the cookies. A cylinder probe with a diameter of 25 mm was used to compress cookies. The parameters of velocity were set at 1.7 mm/s for test speed, 2 mm/s for pretest, and 10 mm/s for after test. Each recipe was evaluated in triplicate and at least three cookies were taken from each recipe and tested.

##### Color Parameters

The chroma a*, chroma b*, and brightness (L*) parameters of the cookie were evaluated using a Miniscan XE (HunterLab) colorimeter with illuminant D65 (daylight) and a 30 mm diameter cell opening.

#### 2.2.5. Morphology of the Dough and Cookies

The morphology of the cookies was evaluated using a scanning electron microscope (SEM) (TM300, Tabletop Microscope Hitachi, Japan). The samples were first frozen using liquid nitrogen, and then freeze-dried (Terroni, São Carlos, Brazil). Finally, the samples were transferred to a microscope where images were recorded at 50×, 100× and 500× magnification and with an accelerating voltage of 15 kV.

#### 2.2.6. Differential Scanning Calorimetry (DSC) Analysis of Cookies 

In order to obtain the local cookie dough temperature increase during baking, a calibrated thermocouple (type T, Ø¼ 0.2 mm) was placed at the dough center of the different cookies and connected to a data acquisition center (Keysight, HP 34972A, Malaysia) [32]. During baking, the temperature profiles of the cookies were acquired every ten seconds, and thereafter plotted versus baking time, to calculate the heating rate. Then, the heating rate was used to simulate the real baking condition using the differential scanning calorimetry (DSC) (DSC 2010, TA instruments, New Castle, DE, USA). Samples (between ~10 mg) were kept at 20 °C for 2 min, then heated up to 120 °C [33].

#### 2.2.7. Sensorial Analysis

The sensory analysis was performed by 100 untrained panelists. Using a 9-point verbal hedonic scale (9 = liked extremely to 1 = disliked extremely), the cookies were evaluated in relation to color, aroma, texture, flavor, and overall acceptability. Prior to performing the test, panelists signed a free and informed consent term. This study was approved by the Ethics Committee of the Faculty of Animal Science and Food Engineering (FZEA/USP) (Process 87793318.9.0000.5422).

#### 2.2.8. Statistical Analyses

The statistical analyses used SAS software (Version 9.2, SAS, Inc., São Paulo, SP, Brazil). Differences between means were determined by the Duncan’s test (95% confidence interval).

## 3. Results and Discussion

### 3.1. Orange By-Product Flour Characterization

#### 3.1.1. Orange Juice and Flour Yield 

First, orange juice was produced, before the flour production. The orange juice yield was 44.8%, in relation to total fruit. This value is close to those reported for orange juice industries, i.e., 44.8% [19]. It is important to underline that orange juice extraction, in this study, was performed manually, at a laboratory scale, using equipment with lower capacities in relation to those used in the industries, which could contribute to a lower juice yield. The OBPF calculated yield was 4.2%, in relation to total fruit.

#### 3.1.2. Chemical Composition 

The OBPF chemical composition is presented in Table 2. The values obtained for water content, protein, fat, ash, dietary fiber, and glycidic fraction were 10.38 ± 0.36 g/100 g of dry matter (DM), 5.94 ± 0.1 g/100 g (DM), 0.33 ± 0.1 g/100 g (DM), 2.72 ± 0.02 g/100 g (DM), 73.61 g/100 g (DM), and 80.63 g/100 g (DM), respectively. These results, on the one hand, are in agreement with those reported by Larrea, Chang, and Martínez Bustos [34] and O’Shea et al. [16], for extruded orange pulp and orange pomace, respectively.

On the other hand, the OBPF produced in this study presented a high level of total dietary fiber, i.e., 73.61% (DM), which is higher than that reported by O’Shea et al. [16], i.e., 40.47%. A high ash value, i.e., 2.72% (DM), indicates that OBPF had a high mineral content. The differences in chemical composition are mainly linked to different cultivars, growing conditions, and also to fruit maturation [10].

#### 3.1.3. Total Phenolic Content

The total phenolic (TP) content of OBPF was 534 ± 30 mg gallic acid equivalent (GAE)/100 g of DM (Table 2). Escobedo-Avellaneda et al. [35] evaluated the TP level in a whole lyophilized orange (juice, flavedo, and albedo), and the result reported was 650 ± 90 mg GAE/100 g of dry matter. Although the result obtained by such authors was higher, it is important to observe that those authors used the entire fruit versus only the albedo as in the present study. Furthermore, the OBPF was submitted to the drying process for 24 h, which could enhance the loss of phenolic compounds.

Danesi et al. [36] reported a lower value (450 ± 30 mg AGE/100 g of dry sample) for the TP content of the same sample type (albedo) in the orange by-products, which could be related to the way the fruit was grown and the heterogeneity of the material. On the basis of a comparison with the literature, it can be observed that the value found for the OBPF was higher than the values reported for flours obtained from other fruit peel. Infante et al. [37] reported results of TP content for pineapple, mango, passion fruit, and cashew residue of 240 ± 6, 450 ± 26, 343 ± 24, and 1067 ± 10 mg GAE/100 g DM, respectively. It was observed that the OBPF content was similar to that reported for mango by-product and higher than that of passion fruit and pineapple. Although the cashew by-product is smaller, the raw material used was processed differently, since the analysis was made using the industrial by-product, not a flour, which is brought to a higher temperature and some phenolic compounds can be degraded. TP content can be degraded or transformed by exposure to high temperatures for an extended period of time [38].

#### 3.1.4. Antioxidant Potential

The results of antioxidant potential measured by FRAP and ORAC assays are shown in Table 2. The antioxidant potential is related to the TP content, thus, the higher the concentrations of the TP content in the flour, the higher the antioxidant potential of this flour. The values reported here for the antioxidant potential measured by the FRAP method for the OBPF are lower than those reported for the orange by-product (albedo) for juice production without the peel (49.7 ± 0.3 µmolTEq/g of DM [36]), apple bagasse (14.4 ± 0.3 mg TEq/g of DM [39]), papaya peel (25 ± 1 µmolTEq/g of DM [40]), and blueberry by-product after extraction of the juice and anthocyanins (39.8 ± 1.1 µmolTEq/g of dry weight).

The values determined for the antioxidant potential by the ORAC method corroborate that determined by Escobedo-Avellaneda et al. [35] (11,953 ± 538 µmolTEq/100 g of DM) in a study of phytochemicals and the antioxidant potential of several orange parts. Chen et al. [41], in a characterization study of flavonoids and its association with the antioxidant potential of orange peel extracts, determined an antioxidant potential of 1117 ± 3 µmolTEq/g of DM, by ORAC assay. This result was higher than the results reported by this study. However, this fact can be explained by the degradation of the active compounds by exposure to high temperatures for an extended period of time [38]. In a study by Chen et al. [41], the orange by-products were dried at 120 °C for 3 h, while in the present study OBPF was dried for 24 h.

#### 3.1.5. Phenolic Profile

Table 3 summarizes the identified phenolic compounds obtained from the UPLC-MS-MS analyses of the orange by-product flour. Some of these compounds, such as quercetin 3-O-rutinoside, have biological relevance to the metabolism of carbohydrate and lipids, and therefore could attenuate some chronic diseases, i.e., hyperglycemia, dyslipidemia, and insulin resistance. Furthermore, dietary plant polyphenols and polyphenol-rich products can modulate oxidative stress, and therefore the pathways sensitive to inflammatory processes [42]. There were no studies in the literature related to the profile of phenolic compounds in flour produced from the residue of the orange juice process. Thus, the results presented once again show important nutritional attributes in the material produced from the industrial by-product.

#### 3.1.6. Hydration Properties

The values obtained for the water absorption index (WAI), the oil absorption index (OAI), and the water solubility index (WSI) of the OBPF were 13.28 ± 0.50 g/g (DM), 2.78 ± 0.02 g (DM), and 9.94%, respectively (Table 2).

The high level of the water absorption index observed for the OBPF (Table 2) was expected, since the OBPF presented high levels of total dietary fiber and the WAI was usually correlated to the quantity and type of fiber [43]. Figuerola et al. [44] reported values between 1.62 and 2.26 g/g (DM) for different fiber concentrates from grapefruit, lemon, orange, and apple. According to Lecumberri et al. [43], the WAI values of 4.76, 0.71, and 5.53 were described for cocoa, cellulose, and carto fiber, respectively. Furthermore, a comparison of the WAI for OBPF (13.28 g/g), and guar gum (17.72 g/g) [45] which normally is taken as a reference, with wheat flour (1.70 g/g to1.90 g/g) [46], emphasizes the high water absorption capacity of the OBPF.

In relation to oil binding capacity, the OBPF presented an OAI of 2.78 g/g (Table 2). The value observed in this study was close to those reported by Figuerola et al. [44], i.e., 1.81 g/g (DM) for “Valencia” cultivar orange residues. The flour nature or thickness surface of the particles influences the OAI [45], and therefore it could be proposed that the OBPF particles present a higher surface which can therefore enhance its ability to adsorb and fix oil components as compared with those of Figuerola et al. [44].

#### 3.1.7. Microstructure Properties

The SEM images of the OBPF (Table 2) under close magnification suggest the presence of fibrous structures with no starch granules. O’Shea et al. [16] also observed a fibrous, laminar structure, for orange pomace by-products by SEM analyses images. Furthermore, they suggested that the lamellar fibrous structure of the particles could represent the insoluble cellulosic fiber, which probably originated from cell wall material.

### 3.2. Cookie Characterization

Figure 1 shows photographs of the cookies formulated using different levels of wheat flour substitution with orange by-product flour (OBPF.) The photos were taken in the following two ways: an image of the entire cookie from the top (Figure 1(a1–e1)) and an image of the cookie when cut in half (cross section) (Figure 1(a2–e2)).

The cookie images show that the cookies produced with the OBPF did not show important visual differences from control cookies produced with only wheat flour (Figure 1). Furthermore, the cross-section images show qualitative differences in alveoli diameter with increased OBPF concentration, i.e., smaller alveoli were observed. This aspect is better discussed in the analysis of cookie microstructure (Section 3.2.2).

#### 3.2.1. Physical Properties Specific Volume, Hardness, and Color Parameters

Table 4 presents the results of physical properties obtained for the different cookies produced with the OBPF. In general, the higher the OBPF concentration, the lower the diameter, the height, and the specific volume of the cookies. However, no significant effect was observed for the spread factor. Larrea et al. [47] observed a similar effect in cookies produced with extruded orange pulp. According to these authors, the pectin content enhanced water absorption, competing for the free water of dough biscuits, and therefore reduced its expansion.

The higher the OBPF concentration, from 0% to 20%, the lower the specific volume of cookies, from 1.56 mL^3^·g^−1^ to 1.38 mL^3^·g^−1^. Kohajdová et al [48] observed the same behavior for the specific volume of biscuits produced with dietary fiber from orange and lemon. The authors explained that the interaction of fiber and gluten could reduce the capacity of dough to retain air, which could explain the decrease in specific volume.

The hardness values of cookies increased significantly due to the increase of the OBPF concentration. The samples with 0%, 5% and 10% OBPF did not differ from each other, however, samples with 15% and 20% of OBPF were significantly harder. A similar effect on biscuits with increased extruded orange pulp was reported by Larrea et al. [47] This increase can be related to the hardness of the fibers in this flour, as well as the OBPF, as the lower hardness is the standard control formulation.

In relation to the color parameters (Table 4), it was observed that the luminosity (L*) was significantly reduced with an increase of OBPF in the cookie formulation. This effect could be expected because the OBPF is darker as compared with the wheat flour (standard control formulation). In Baumgartner et al [49], the incorporation of oat bran altered the color of the cookies produced, especially causing a darker color as its percentage was increased. Nevertheless, no significant differences in a* and b* values were found between control samples and samples enriched with OBPF.

#### 3.2.2. Microstructure

Figure 2 presents the images obtained by SEM (with different magnifications 50×, 100×, and 500×) of the cookies produced with different concentrations of OBPF.

Images of the control cookies formulation showed the presence of an alveolar structure, or a pore, in the matrix. This pore could be attributed to the wheat flour and to a well-developed gluten network, as in this case, dough was constituted with only wheat flour. The higher the level of wheat flour substitution with OBPF, the lower the pore size/presence. The OBPF had a higher level of fiber in its composition, therefore, the increase of OBPF could destabilize gluten development, and could produce a more compacted dough. Compared with a study by Blanco et al. [50] in which they evaluated the effect of different dietary fiber inclusion in cookie dough, and related that the oat fiber presented irregular, squamous, and heterogeneous particles, the results were similar to those observed for cookies with a higher concentration of OBPF.

#### 3.2.3. Differential Scanning Calorimetry (DSC)

On the basis of the DSC analyses, in general, no significant differences were observed for the endothermic parameters obtained from cookies produced by different levels of wheat flour substitution with OBPF (Table 5).

Kulp et al. [51] reported similar results in a DSC cookie study, suggesting that starch in dough cookies remained unchanged during baking. In addition, the values of transition temperatures observed in the present study are in agreement, and very close to those reported by Chevallier et al. [33] for cookies produced with different ingredients. These authors reported three thermal transition temperatures, at 94, 114 and 130 °C, and related them to starch melting.

Pareyt and Delcour [30] stated that sugar could act simultaneously as a plasticizer, by decreasing and narrowing the gelatinization temperature range, and as an anti-plasticizer, by raising the gelatinization temperature. The authors suggested that the starch did not gelatinize because the percentage of water in the sample was probably too low for gelatinization. Thus, it could be supposed that the increase of OBPF had the same effect, i.e., no significant difference in the phenomes of starch transition, once the sucrose level used was the same for dough cookies.

#### 3.2.4. Sensorial Analyses

Table 4 shows sensory parameters of cookies produced with OBPF. The substitution of wheat flour with OBPF significantly influenced all sensorial attributes. Incorporation of OBPF at 10% levels significantly increased color, aroma, texture, flavor and overall acceptability. There were no significant differences between the control cookies and those prepared with 5 g/100 g OBPF. Furthermore, cookies with 10% of OBPF presented better scores than the control for all attributes.

Kohajdová et al. [48] also observed the effect of lower in overall acceptance of biscuits with concentrations of orange and lemon fibers, from a concentration higher than 10%.

## 4. Conclusions

An adequate process was proposed for developing the orange juice by-product in the food industry, representing a strategy for industries once the orange juice by-product produced represents more than 4% of total fruit. In addition, orange juice by-product flour represents a potential approach to improve “fruit” intake and dietary fiber. Moreover, its complete characterization, in terms of not only its composition, but also in relation to functional properties, before its application in real and easy to manufacture food, demonstrated the potential of the OBPF in food industry.

Notwithstanding the exposure to high temperatures during the drying, the OBPF presented high concentration of total phenolic and a high antioxidant potential. In addition, high WAI values are favorable for the use of OBPF in bakery products. Cookies produced with OBPF were well accepted by tasters. All of this is in agreement with the FAO recommendations.

An adequate destination of by-products from industrial fruit processing represents a sustainable and key strategy, since the fruit groups are most responsible for food waste in all supply chain stages. The strategy proposed in this paper could represent a competitiveness aspect for the orange juice industry, once significant costs related to waste treatment are lowered and, at same time, there is a high value-added by-product. In addition, the by-product recycling cycle forms part of the current sustainable development and environmental protection.

## Figures and Tables

**Figure 1 foods-09-00593-f001:**
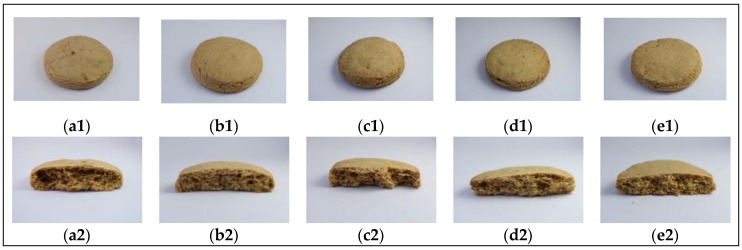
Photographs of cookies formulated using different levels of wheat flour substitution with orange by-product flour (OBPF). (**a**) control; (**b**) 5%; (**c**) 10%; (**d**) 15%; and (**e**) 20%; (**1**) upper image; and (**2**) cross-sectional image.

**Figure 2 foods-09-00593-f002:**
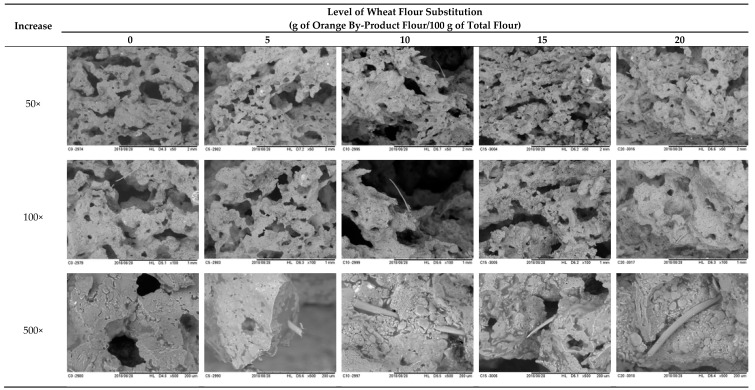
Images obtained from scanning electron microscopy (SEM) analysis of cookies produced by wheat flour substitution with orange by-product flour (OBPF), at magnifications of 50×, 100× and 500×.

**Table 1 foods-09-00593-t001:** Cookie dough composition for different levels of wheat flour substitution with orange by-product flour (OBPF).

Ingredients (g)	Level of Wheat Flour Substitution (g of Orange By-Product Flour/100 g of Total Flour)				
	Control	5	10	15	20
Wheat flour	238.50	226.57	214.65	202.73	190.80
OBPF	0	11.93	23.85	35.77	47.70
Margarine	43.5	43.5	43.5	43.5	43.5
Brown sugar	86	86	86	86	86
Sugar	86	86	86	86	86
Water	8	8	8	8	8
Chemical leavener	5	5	5	5	5

**Table 2 foods-09-00593-t002:** Results from the different analysis of the orange by-product flour produced.

	Analysis/Component	Value
Chemical composition	Water content (g/100 g of DM *)	10.38 ± 0.36
	Protein (g/100 g of DM)	5.94 ± 0.1
	Fat (g/100 g of DM)	0.33 ± 0.1
	Ash (g/100 g of DM)	2.72 ± 0.02
	Dietary fiber (g/100 g of DM)	73.61
	Glycidic fraction (g/100 g of DM)	80.63
Antioxidant properties	Phenolic compound (mg GAE/100g of DM) **	534 ± 30
	FRAP (µmolTEq/100 g of DM) ***	93 ± 5
	ORAC (µmolTEq/100 g of DM) ***	11,728 ± 541
Functional properties	WAI (g of water/g of DM)	13.28 ± 0.50
	OAI (g of oil/g of DM)	2.78 ± 0.02
	WSI (%)	9.94
Microstructure	SEM image at 500× magnification	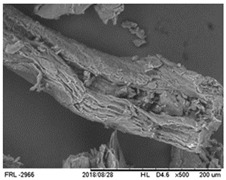

* DM, dry matter; ** GAE, gallic acid equivalent; and *** TE, trolox equivalent.

**Table 3 foods-09-00593-t003:** The HPLC-DAD-ESI-QTOF-MS/MS analysis of phenolic compounds (flavonols) in orange by-product flour, retention times (RT), mass spectrum data.

N°	RT	[M-H]-(*m*/*z*)	MS2 (*m*/*z*)	Compound
1	6.8	609.1467	519/489/429/399/369.0601	Luteolin 6,8-di-C-glucoside
2	7.6	593.1517	473/353/383/353/325.0666/297	Apigenin 6,8-di-C-glycopyranoside (Vicenin-2)
3	7.8	623.1613	503/413/383/355/312.0646	Chrysoeriol 6,8-di-C-glucoside (Stellarin 2)
4	8.2	623.1635	503/413/383/355/312.0659	Chrysoeriol 6,8-di-C-glucoside (Stellarin 2)
5	9.6	563.1427	413/293.0459	-
6	10.0	609.1488	300.0288/151	Quercetin 3-*O*-rutinoside (Rutina)
7	11.7	579.1741	271.0613/151	Naringenin-7-hesperidoside (Naringin)
8	13.0	609.1831	301.0717/164	Hesperetin 7-*O*-rutinoside (Hesperidin)

**Table 4 foods-09-00593-t004:** Physical characteristics and sensorial analysis results of cookies produced by wheat flour substitution with orange by-product flour at different levels.

**Parameter**	**Level of Wheat Flour Substitution** **(g of Orange By-Product Flour/100 g of Total Flour)**				
	**0**	**5**	**10**	**15**	**20**
Diameter (mm)	54.05 ± 0.7 b	53.14 ± 1 ab	53.07 ± 1 ab	53.58 ± 0.8 ab	52.55 ± 0.8 a
Height (mm)	12.63 ± 0.8 b	12.39 ± 0.4 ab	12.15 ± 0.5 ab	11.78 ± 0.6 a	11.82 ± 0.7 a
Spread factor (SF)	4.96 ± 0.3 a	4.29 ± 0.2 a	4.38 ± 0.3 a	4.56 ± 0.3 a	4.46 ± 0.2 a
Specific volume (mL/g)	1.56 ± 0.04 c	1.50 ± 0.1 b	1.55 ± 0.04 c	1.48 ± 0.03 b	1.38 ± 0.03 a
Hardness (N)	101.01 ± 22 a	114.0 ± 20 a	117.0 ± 15 a	176.9 ± 45 b	275.6 ± 37 c
L*	64.19 ± 8 b	61.68 ± 5 ab	60.82 ± 7 ab	57.71 ± 3 a	56.20 ± 4 a
Chroma a*	8.94 ± 0.7 a	10.01 ± 2 a	10.19 ± 2 a	8.93 ± 1 a	9.60 ± 0.9 a
Chroma b*	28.98 ± 2 a	29.63 ± 4 a	29.45 ± 4 a	26.83 ± 0.9 a	27.22 ± 2 a
**Sensorial attribute ***	**0**	**5**	**10**	**15**	**20**
Color	6.42 ± 1.5 ab	6.68 ± 1.6 bc	7.05 ± 1.4 c	6.00 ± 1.6 a	6.65 ± 1.5 bc
Aroma	6.53 ± 1.5 ab	6.82 ± 1.4 bc	6.98 ± 1.4 c	6.14 ± 1.4 a	6.55 ± 1.5 ab
Texture	6.41 ± 1.9 c	6.27 ± 1.8 c	6.48 ± 1.9 c	4.71 ± 2.1a	5.5 ± 2.2 b
Flavor	6.62 ± 1.7 bc	7.04 ± 1.4 cd	7.16 ± 1.5 d	5.81 ± 1.8 a	6.52 ± 1.6 b
Overall acceptability	6.66 ± 1.5bc	6.81 ± 1.4 c	7.07 ± 1.3 c	5.52 ± 1.6 a	6.35 ± 1.5 b

Data expressed as mean ± standard deviation. Values followed by different letter in the same line are significantly different at 95% confidence level. * scale between 1 and 9 (9 = “liked extremely” and 1 = “disliked extremely”).

**Table 5 foods-09-00593-t005:** Mean values obtained from the transition temperatures (Ts, start temperature; T_o_, onset temperature; T_m_, maximum temperature; and T_stop_, final temperature) and the enthalpy of the cookie with different concentrations of orange by-product flour (OBPF).

Level of Wheat Flour Substitution (g of Orange By-Product Flour/100 g of Total Flour)	Temperature (°C)				ΔH (J/g)
	T_s_	T_o_	T_m_	T_stop_	
0	105.16 ± 5.62 a	108.9 ± 7.01 a	131.01 ± 5.29 a	156.71 ± 1.18 a	4.30 ± 0.38 a
5	111.46 ± 6.95 a	118.06 ± 3.03 b	131.81± 3.49 a	158.00 ± 2.56 a	4.18 ± 0.17 a
10	108.23 ± 4.44 a	112.59 ± 3.26 ab	128.14 ± 2.18 a	157.39 ± 2.75 a	5.67 ± 0.25 b
15	110.11 ± 4.57 a	113.92 ± 5.28 ab	130.32 ± 0.67 a	156.14 ± 0.25 a	4.40 ± 1.16 a
20	111.59 ± 0.46 a	116.70 ± 0.55 ab	131.90 ± 0.73 a	157.39 ± 1.78 a	4.11 ± 0.55 a

Data expressed as mean ± standard deviation. Values followed by the same letter in the same line are not significantly different at 95% confidence level.

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
