# Peer review of "From Orange Juice By-Product in the Food Industry to a Functional Ingredient: Application in the Circular Economy"

_foods, 2020, doi:10.3390/foods9050593_

Round 1

Reviewer 1 Report

This is well written and interesting manuscript.

I have only minor comments:

  1. What means chemical yeasts?
  2. Can You compare hydration properties to wheat flour?
  3. What with antioxidant activity of OBPF in baked cookies
  4. I found some minor editorial mistakes:line 157 cromatographicline 263 bracketline 319 byproduct should be by-productTable 3 - please check names of compounds, I see some mistakesline 434 - increases- it should be magnification (the same in the table)447 - snetence should be correctedline 488 - fragment of sentence is missing

Language needs to be checked again, some mistakes happens in the text. 

Reviewer 2 Report

Many articles have been realized on the valorization of orange by-products. You should mention them in the introduction section.

You should also inserted some refernces about the global quantity of orange peels which can be collected from industries. 

Please consider as reference the work of Caldeira et al. (2019), where you can find the potential amount of food by-prodcuts originated by the industry.

Insert also in the conclusion section a discussion on the industrial feasibility of your process and the profitability potential of your application. 
